# Medical students in distress: The impact of gender, race, debt, and disability

Nikhil Rajapuram[1]*, Simone Langness[1], Megan R. Marshall[2], Amanda Sammann[1]

1 Department of Surgery, University of California, San Francisco, San Francisco, CA, United States of America, 2 New York University School of Medicine, New York, NY, United States of America

☯ These authors contributed equally to this work.
* nikhil.rajapuram@ucsf.edu

**Data Availability Statement:** All relevant data are within the Supporting Information files.

**Funding:** NR received a stipend from the UCSF School of Medicine as a research fellow to complete this research. The funders had no role in

## Abstract

### Background

In 2012, over half of US medical students experienced burnout and depression. Since that time, there have been many changes to student demographics, school resources and awareness of burnout in the medical field altogether. New tools are also available to screen for student distress, a condition that correlates with low mental quality-of-life, suicidal ideation and serious thoughts of dropping out. Despite increased attention on wellbeing and improved screening methods, no large-scale studies have evaluated student distress in the modern era of medical education. The objective of this study was to determine the current prevalence of medical student distress and contributing risk factors.

### Methods

Student wellbeing from a national cohort of US medical students was measured with an electronic survey in a prospective, observational survey study from 2019–2020. Medical student distress was defined as a Medical Student Wellbeing Index (MS-WBI) of ≥4. Demographic details including age, race, gender, marital status, disability, desired specialty, and debt burden were evaluated in a multivariate logistic regression model to determine possible risk factors for the development of distress.

### Results

A total of 3,162 students responded to the survey, representing 110 unique medical schools. Of these respondents, 52.9% met criteria for distress and 22% had either taken or considered taking a leave of absence for personal wellbeing. Independent risk factors for distress included involvement in the clinical phase of medical school (OR 1.37); non-male gender (OR 1.6); debt burden >$20,000 (OR 1.37), >$100,000 (OR 1.81), and >$300,000 (OR 1.96); and disability status (OR 1.84).

### Conclusions

Medical student wellbeing remains poor in the modern era of medical education despite increased attention to wellbeing and increased availability of wellbeing resources. Disability

study design, data collection and analysis, decision to publish, or preparation of the manuscript.

**Competing interests:** The authors have declared that no competing interests exist.

status is a novel risk factor for distress identified in this study. The persistence of previously identified risk factors such as non-male gender, debt burden and clinical phase of school suggest that efforts to curb medical student distress have been inadequate to date.

## Introduction

Medical students experience alarmingly high rates of distress. Distress can manifest as depression, burnout, alcohol abuse, suicidal ideation or simply a desire to leave the field of medicine altogether [1–5]. It is estimated that medical students are 40% more likely to be burned out and 20% more likely to be depressed compared to age-matched controls [6], despite having higher mental and physical quality-of-life scores at matriculation [7].

In 2012, the prevalence of burnout, depression, and suicidal ideation amongst US medical students was 55.9%, 58.2% and 9.4%, respectively [8]. Since that time, medical education has undergone considerable change. Medical school matriculants are more diverse than ever before [9, 10]. Females now account for over half the current medical students in the country [10] and students who identify as having a disability have increased five-fold over the last five years [11]. African American, Hispanic and Asian American student enrollment has also increased sharply [9]. Current medical students are shaped by technology and utilize innovative methods to learn medical content [12]. They also face new challenges in school, such as catastrophic debt burden [13, 14], highly competitive residency positions [15] and the need to keep pace with an exponential increase in medical knowledge.

Medical schools are well aware of the negative consequences that medical education can have on student wellbeing. In an effort to curb medical student distress, some schools have begun offering pass/fail-grading systems [16] and novel curricula designed to shorten the preclinical phase [17]. Schools have also invested considerably in wellbeing resources [18, 19] as awareness and dialogue about burnout in the medical field has become inescapable [20, 21].

Lastly, the scientific and psychological understanding of personal wellbeing has grown tremendously [22]. Wellbeing is now considered along multiple domains (i.e., emotional, physical, social, financial, etc) [23]; providing a more holistic view of one's state of being. The breadth of research about personal wellbeing has led to the development and validation of novel tools to study this subject. In 2010, Dyrbye et al. created the Medical Student Wellbeing Index (MS-WBI), a comprehensive and simple questionnaire to screen for student distress utilizing questions from the Maslach Burnout Index, the PRIME-MD depression test and the Short Form-8 mental and physical quality of life screening [24]. They found that an MS-WBI score ≥4 was associated with severe distress, correlating with low mental quality-of-life, recent suicidal ideation and/or serious thoughts of dropping out of school, with a sensitivity and specificity of 90% [25]. Using this definition, the prevalence of severe distress was 31.7% [26]. Since that time, the MS-WBI has been adapted to screen for distress in resident [27] and attending physicians [28].

Despite an increased understanding of the complexities of personal and professional wellbeing, a change in medical student demographics, and an increased awareness of burnout in medical schools, there have been no national studies in the last eight years measuring medical student wellbeing. Recent literature has focused on small sample sizes and single institutions. The purpose of this study was to quantify US medical student wellbeing in the modern era of medical education.

## Materials and methods

### Study overview

This was a prospective observational study to evaluate the prevalence of medical student distress and to determine contributing risk factors. Medical students were surveyed via an electronic questionnaire after institutional review board (IRB) approval was obtained from the University of California, San Francisco. Informed consent was obtained electronically prior to survey participation. The quantitative data in this study is part of a larger research project on medical student wellbeing.

### Survey development

A novel instrument, the Medical Student Wellbeing Survey (MSWS), was developed to evaluate various aspects of medical student wellbeing. This 30-question survey assessed student demographics, specialty consideration, and stressors (S1 Appendix) and included a combination of validated and novel survey questions. The validated MS-WBI questionnaire was administered within the survey with permission from the authors. Novel questions were created based on foundations established from previously validated wellbeing survey methods [29–31]. Changes in students' physical, emotional, and social domains of wellbeing since entering medical school were assessed on a five-point Likert scale. Students were asked to designate whether they identified as having a disability and/or chronic illness. Additionally, respondents were asked about their frequency of participation in activities known to be protective against distress and burnout, such as exercise, meditation/mindfulness practices, healthy eating, hobby participation, and connecting with a loved one [32–35]. Frequency of actual participation in protective activities was compared against students' self-reported ideal participation frequency.

The survey was piloted with a cohort of 10 medical students from various institutions and demographics over two testing cycles prior to administration. During the first cycle, time to survey completion was recorded for each participant. After completing the survey, each participant was interviewed for question clarity and survey content and completeness. Feedback obtained from the first pilot cycle was incorporated, and the survey was readministered to the 10 participants. Upon completion of the second pilot cycle, all interview participants reported that the survey questions were easy to comprehend and adequately addressed the topic of medical student wellbeing.

### Participant selection and recruitment

The MSWS was administered from September 2019 to February 2020. Participants were included if they were actively enrolled in an accredited US or Caribbean medical school during the survey administration time period. For survey distribution, we first contacted the Association of American Medical Colleges (AAMC). However, the AAMC administers a national survey that targets second-year and graduating medical students only, does not include the MS-WBI, and would not allow for individual response tracking. We also contacted several Medical School Deans who voiced concern that respondents were asked to identify their medical school. Ultimately, to obtain national survey distribution, we sent a recruitment email to a medical student liaison (MSL) at every medical school in the US and Caribbean, identified through the AAMC medical student representative listserv. In total, seventy-four MSLs responded to our request with interest to distribute the survey. MSLs were provided a copy of the study's IRB approval and study materials (email recruitment letter and survey link). MSLs then redistributed the study materials to their own classmates. Access to the survey was also

available through social media platforms on Twitter and Facebook. Participation in the survey was voluntary and all students were advised to take the survey once. The need to administer the survey through MSLs precluded the ability to determine the response rate.

### Student characterization and geomapping

Respondents were asked to identify their phase in medical school. "Pre-clinical," "Clinical," and "Post-Clinical" were defined as time spent prior to, during, or having completed core clinical clerkships, respectively. "Gap" was defined as dedicated time away from clinical work for research, additional degrees and/or to take a leave of absence. Respondents identified their medical school, which was further categorized by state and region, as defined by US Census Bureau [36]. Severe distress rate per state was mapped using the online software DISPLAYR. States with <10 respondents were excluded from severe distress rate-mapping.

### Specialty competition

Survey participants were asked to select their intended specialty choice and confidence in the stated specialty choice. Specialties were then categorized by competitiveness based on 2018 National Resident Match Program data. High and Low competition was defined as an average United States Medical Licensing Examination (USMLE) Step 1 score of >240 or <230, respectively or if >18% or <4% of applicants were unmatched, respectively. Moderate competition was defined as any specialty not meeting criteria for either High or Low competitiveness.

### Medical student stressors

Survey respondents were asked about the effect of various factors on their stress level using a five-point Likert scale. Candidate variables were identified in the literature as contributing to student distress which included year in medical school, age, gender, marital status, debt burden, underrepresented minority (URM) status, disability status, specialty competitiveness, and confidence in specialty [5, 26, 37]. Mean values of scores were displayed and used to derive a stressor ranking list.

### Risk factor modeling

A multivariable logistic regression model was developed to determine risk factors for severe distress. The data was randomly split (7:3) into training and validation cohorts to allow for model building and testing, respectively. The cohorts were well matched for severe distress and all candidate variables with an absolute value of the standardized difference of <0.2. Missing data was present in <5% of all respondents in both the training and validation cohort.

Using least absolute shrinkage and selection operator (LASSO) and net elastic regression, a model was created to predict the presence of severe distress in medical students from candidate variables. The tuning parameter associated with the smallest cross-validated error was used. The multivariable regression model had an adequate fit with a Pearson chi-square goodness-of-fit test equal to 0.2. The model was then tested on the validation data cohort with similar receiver operating characteristic curves. Additionally, observed and predicted probability for severe distress from the validation cohort was closely matched.

All statistical tests were two-sided and p <0.05 was considered significant. Statistical analyses were performed using SAS version 9.4 and R version 3.6.1. The "glmnet" package in R was used to fit the elastic net and LASSO penalized regression models.

## Results

### Survey respondents

A total of 3,162 students responded to the survey. Respondents represented 110 unique medical schools throughout the US, Caribbean and US territories. Students from medical schools in the Caribbean and/or US territory represented <0.1% of all responses. There was an even distribution of respondents from medical school phases with 52.0% in the pre-clinical phase and 42.8% in the clinical/post-clinical phase (Table 1). A small proportion (5.2%) of respondents identified as being in a "gap" year. Most respondents were female (64.5%), between the ages of 22–27 (79.8%) and never married (87.2%). According to the definitions set forth by the AAMC, 11.4% of respondents were characterized as URM, including Black / African American, Hispanic / Latinx and Native American [38]. Nine percent of respondents identified as having a disability or chronic illness.

### Respondent geographic distribution

Respondents represented medical schools from 33 states with the largest percentage of respondents from New York (15.6%), Pennsylvania (15.3%) and California (12.7%). The average number of respondents per school was 28.5 (range 1–224).

### Severe distress in US medical students

The average MS-WBI score was 3.56 ± 1.95. A total of 1,646 respondents (52.9%) had an MS-WBI score ≥4, indicating severe distress (Fig 1A). Respondents reported a decline in physical (60.9%) emotional (66.3%) and social (56.9%) wellbeing since starting medical school (Fig 1B). They also reported a significant gap (>3 days) between their ideal and real participation in mindfulness practices (22%), hobbies (15%) and exercise (12%) (Fig 1C). A mild gap (1–3 days) between ideal and real participation was reported for hobbies (58%), exercise (58%) and healthy eating (57%). The percentage of respondents who reported taking a leave of absence from medical school for their personal wellbeing was 3.8% and 17.5% had considered a leave of absence (Fig 1D). Severe distress rates ranged from 39% (Massachusetts) to 72% (Wisconsin) (Fig 2). The average rate of severe distress by state was 53.8% ± 9.6%.

### Risk factors for severe distress

Results of the multivariable regression model are listed in Table 2. Controlling for all other variables in the model, severe distress was more likely for students in their clinical phase (OR 1.4, 95% CI 1.1–1.8, p = 0.02) and those in a gap year (OR 1.8, 95% CI 1.1–2.8, p = 0.01) compared to students in their pre-clinical coursework phase. Additional variables significantly associated with severe distress included non-male gender, disability, and higher debt burden. All other associations in the model were not significant.

### Medical student stressors

Top stressors for all students were national board exams (1.55 ± 0.71), grades (1.33 ± 0.73), uncertainty about the future (1.22 ± 0.77), isolation from family and friends (1.16 ± 0.76) and lack of control over one's schedule (1.07 ± 0.8) (Fig 3A). Stressor ranking did not vary substantially according to medical school year, gender, debt burden, race or disability status (Fig 3B).

## Discussion

Medical student wellbeing remains poor in the modern era of medical education. Our study found that over half of the students we surveyed experienced severe distress, a rate 20% higher

**Table 1. Demographic characteristics of the study cohort.**

| | Pre-Clinical | Clinical | Post-Clinical | Other | Grand Total |
|---|---|---|---|---|---|
| N (%) | 1644 (52) | 643 (20) | 710 (22) | 165 (5) | 3162 |
| **Age, N (%)** | | | | | |
| <21 | 46 (1) | 1 (0) | 0 (0) | 0 (0) | 47 (2) |
| 22–27 | 1388 (45) | 490 (16) | 474 (15) | 111 (4) | 2463 (79) |
| ≥28 | 179 (6) | 135 (4) | 220 (7) | 42 (1) | 576 (18) |
| **Gender, N (%)** | | | | | |
| Male | 533 (17) | 233 (7) | 223 (7) | 54 (2) | 1043 (33) |
| Non-Male* | 1073 (34) | 390 (13) | 469 (15) | 96 (3) | 2028 (65) |
| **Marital Status, N (%)** | | | | | |
| Never Married | 1470 (47) | 535 (17) | 554 (18) | 131 (4) | 2690 (86) |
| Married | 136 (4) | 83 (3) | 134 (4) | 21 (1) | 374 (12) |
| Divorced / Widowed | 8 (0) | 7 (0) | 5 (0) | 1 (0) | 21 (1) |
| **Debt Burden, N (%)** | | | | | |
| <20K | 526 (17) | 155 (5) | 188 (6) | 54 (2) | 923 (30) |
| 20-100K | 681 (22) | 121 (4) | 111 (4) | 27 (1) | 940 (30) |
| 100-300K | 291 (9) | 292 (9) | 278 (9) | 50 (2) | 911 (29) |
| >300K | 10 (0) | 25 (1) | 89 (3) | 9 (0) | 133 (4) |
| Not Sure | 56 (2) | 16 (1) | 9 (0) | 8 (0) | 89 (3) |
| **Race, N (%)** | | | | | |
| Asian | 323 (10) | 111 (4) | 131 (4) | 31 (1) | 596 (19) |
| Black / African American | 76 (2) | 29 (1) | 24 (1) | 13 (0) | 142 (5) |
| White | 929 (30) | 376 (12) | 397 (13) | 75 (2) | 1777 (57) |
| Hispanic / Latino | 78 (3) | 35 (1) | 60 (2) | 13 (0) | 186 (6) |
| Multiracial | 161 (5) | 53 (2) | 48 (2) | 12 (0) | 274 (9) |
| Other | 16 (0) | 7 (0) | 10 (0) | 2 (0) | 34 (1) |
| **URM, N (%)** | | | | | |
| Yes | 162 (5) | 66 (2) | 88 (3) | 26 (1) | 342 (11) |
| No | 1421 (46) | 545 (18) | 581 (19) | 120 (4) | 2667 (86) |
| **Disability, N (%)** | | | | | |
| Yes | 156 (5) | 49 (2) | 49 (2) | 23 (1) | 277 (9) |
| No | 1435 (46) | 559 (18) | 636 (20) | 128 (4) | 2758 (89) |
| **Desired Specialty Competitiveness, N (%)** | | | | | |
| Low | 695 (22) | 309 (10) | 387 (12) | 73 (2) | 1464 (47) |
| Moderate | 600 (19) | 214 (7) | 219 (7) | 44 (1) | 1077 (35) |
| High | 289 (9) | 92 (3) | 78 (3) | 31 (1) | 490 (16) |
| **Confidence in Specialty, N (%)** | | | | | |
| Low | 479 (15) | 84 (3) | 34 (1) | 18 (1) | 615 (20) |
| Moderate | 474 (15) | 159 (5) | 34 (1) | 24 (1) | 691 (22) |
| High | 660 (21) | 382 (12) | 625 (20) | 110 (4) | 1777 (57) |

Abbreviations: URM = underrepresented minority

* Includes Female, Transgender, Other, Prefer Not to Say

than that identified in a national cohort from 2007 [25]. Furthermore, we found that one in five students had either taken or considered taking time off from medical school specifically for their personal wellbeing. The high rate of medical student distress identified in this study suggests that current efforts to improve medical student wellbeing are inadequate.

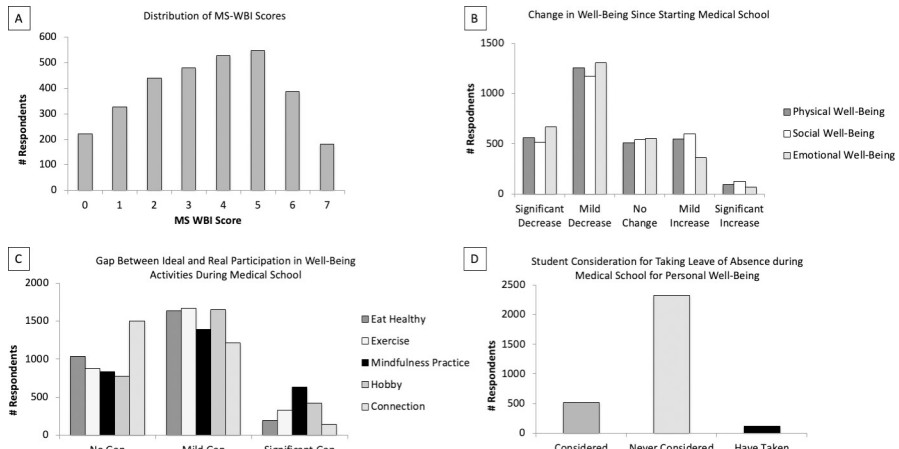

**Fig 1. Medical Student Wellbeing Survey results by component.**

## Estimation of distress

To our knowledge, this is the first study to utilize the MS-WBI in a national sample of medical students since the tool was first validated. While we primarily evaluated MS-WBI as a binary index of severe distress, additional insight can be obtained through a more thorough evaluation of specific scores. Hansell et al. estimated burnout by measuring affirmative responses to the following MS-WBI questions: *1) Do you feel burned out from medical school*? or *2) Do you worry medical school is hardening you emotionally*? [5] Utilizing the same definition in our study population would have yielded a burnout rate of 76%. An affirmative answer to the PRI-ME-MD question within the MS-WBI (*During the past month, have you often been bothered by feeling down, depressed, or hopeless*?) would have resulted in a 49.3% rate of depression within our study population [24]. Furthermore, escalating MS-WBI scores are correlated with worsening wellbeing. Dyrbye et al. found that the likelihood ratio (and probability) for low mental quality-of-life, serious thoughts of dropping out, and suicidal ideation for students with an MS-WBI score of $\geq 6$ were 12.1 (88.9%), 7.2 (44%) and 3.8 (29.6%), respectively [26]. These

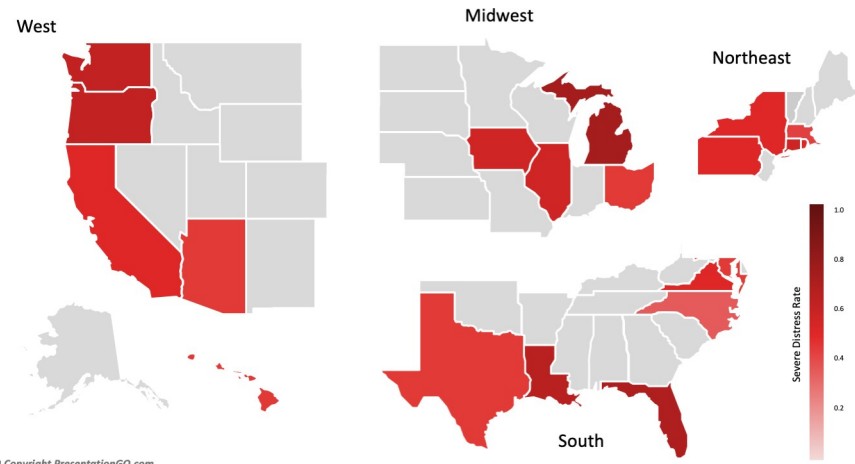

**Fig 2. Rate of severe distress amongst medical students by state & region.** Darker color corresponds to higher rates of Severe Distress as defined by MS-WBI.

**Table 2. Predictors of severe distress in medical students from multivariable logistic regression model.**

|  | Odds Ratio (95% CI) | *P*-value |
|---|---|---|
| **MS Year (vs. Preclinical)** |  |  |
| Clinical | 1.37 (1.06–1.77) | *.02** |
| Post-Clinical | 0.80 (0.61–1.06) | .12 |
| Other | 1.78 (1.12–2.84) | *.01** |
| **Age (vs. 22–27)** |  |  |
| <21 | 0.69 (0.32–1.45) | .33 |
| ≥28 | 0.99 (0.76–1.29) | .93 |
| **Gender (vs. Male)** |  |  |
| Non-Male | 1.60 (1.31–1.95) | *< .001** |
| **Marital Status (vs. Never Married)** |  |  |
| Married | 1.18 (0.88–1.59) | .28 |
| Divorced / Widowed | 1.04 (0.32–3.37) | .94 |
| **Debt Burden (vs. <$20K)** |  |  |
| 20-100K | 1.37 (1.08–1.72) | *.009** |
| 100-300K | 1.81 (1.42–2.31) | *< .001** |
| >300K | 1.96 (1.19–3.24) | *.009** |
| Not Sure | 1.22 (0.70–2.13) | .48 |
| **URM (vs. No)** |  |  |
| Yes | 1.11 (0.83–1.32) | .49 |
| **Disability (vs. No)** |  |  |
| Yes | 1.84 (1.32–2.57) | *< .001** |
| **Desired Specialty Competitiveness (vs. Low)** |  |  |
| Moderate | 1.19 (0.97–1.46) | .10 |
| High | 1.16 (0.89–1.51) | .28 |
| **Confidence in Specialty Choice (vs. Low)** |  |  |
| Moderate | 1.07 (0.84–1.36) | .59 |
| High | 0.82 (0.63–1.07) | .14 |

* Statistically significant at p < .05

scores increased to 18.5 (92.5%), 11.0 (55%) and 14.3 (61.4%) when MS-WBI score was 7. In our study, 388 respondents (12.5%) had an MS-WBI score of 6 and 131 respondents (5.8%) had a score of 7. Importantly, one student with a score of 6 in our study population died by suicide during the study period.

## Individual versus structural interventions

Current efforts to curb medical student distress have overwhelmingly focused on individuals rather than the underlying structure of medical education [37, 39, 40]. In our data, board exams, grades and uncertainty about the future were the most common student stressors regardless of individual characteristics. Programs such as resiliency training, mindfulness, and self-care education have all aimed to give students capacity to endure the challenges of medical school. Yet, little has been done to address increased competition for residency positions, expectation of mastery over expanding medical content or decreased autonomy and responsibility that medical students face on clinical rotations. Our data suggest that changes to the underlying structural components of medical education are likely to yield more substantial results to improve medical student wellbeing than continued focus on individuals.

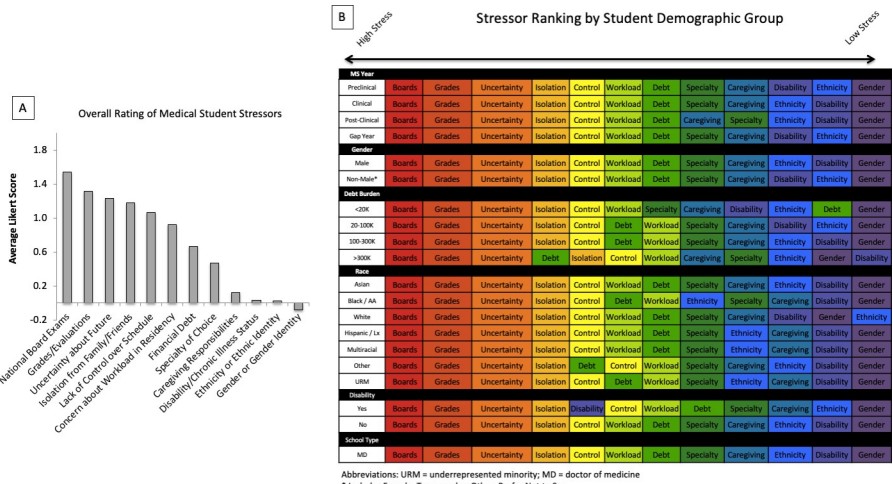

**Fig 3. Key stressors for medical students.** A: A 5-point Likert scale was used with positive scores corresponding increased stress due to specific stressor. B: Abbreviations: AA = African-American; Lx = Latinx.

Efforts to address risk factors known to contribute to poor medical student wellbeing have largely been ignored. The clinical phase of medical school as illustrated by our, and other studies is a particularly vulnerable time for medical students, yet most interventions to date focus on the preclinical phase [16, 41, 42]. Furthermore, our study and preceding studies found debt burden to significantly affect distress, yet schools have become more expensive with time. According to the National Center for Education Statistics, the total loan debt taken on by medical students at the time of graduation has more than doubled over the last decade from an average cost of $108,000 in 2003 to $223,000 in 2016 after adjusting for inflation [14]. Lastly, female or genders other than cisgender male are at a disproportionate risk of distress while also having higher representation in medical schools than ever before [10].

## New risk factors

Our study identified disability status as a novel risk factor for the development of severe distress. We found that nearly 10% of survey respondents identified as having a disability and that disability status nearly doubled the risk of severe distress (OR 1.81). Self-reporting of disability in medical school has risen substantially over the last three years, specifically for psychological and chronic health issues. The prevalence of self-identified disability in our study is higher than previously cited (4.6%), which may indicate a higher willingness of students to disclose their disability if done with anonymity as in our study [43]. Little is currently known about the additional stressors faced by students with disability and to what degree accommodations from medical schools are being offered and utilized. Understanding this topic more thoroughly may prove pivotal for guiding medical schools in designing interventions to support this vulnerable group.

## Study limitations

A major limitation to this study is the inability to determine a response rate of students. Our survey was conducted as part of a larger study on medical student wellbeing that required the ability to track individual responses and medical school identity. Survey distribution through national organizations or through selective school participation proved unable to fulfill this need. As such, we pursued an alternate survey distribution method for which calculating a

response rate was not possible. Assuming the survey was distributed to every US allopathic medical student, the response rate would be 8.7% [44].

A second limitation is the potential for response bias in our study. Respondents may not accurately reflect the greater medical student cohort and there was a disproportion of female survey respondents, which may result in lower than expected rates of wellbeing. Shanafelt et al. investigated the topic of response bias amongst US physicians who participated in a survey on burnout and wellbeing. They compared initial survey respondents to respondents who required intensive follow-up and solicitation to participate. There was a higher proportion of females in the initial survey respondent group but no difference in respondent characteristics, burnout prevalence or wellbeing scores between the groups [45].

## Conclusion

Current medical students are facing poor personal wellbeing at record levels. The proportion of students experiencing adverse effects from medical school including distress, burnout, and depression is rising despite increased attention to and discussion on the subject over the last decade. While some components of poor wellbeing may be individual, this alarming trend points to more systemic problems within medical education. If the medical education field is unable to create meaningful change toward improving medical student wellbeing, we risk an increasing number of talented students abandoning or never pursuing a career in medicine, a growing epidemic of distressed medical students, or worse, further irreplaceable loss of life from death by suicide.

## Supporting information

**S1 Appendix. Medical Student Well-being Survey.**
(DOCX)

**S1 Dataset.**
(XLSX)

## Acknowledgments

We thank the many medical students who assisted in the distribution of the Medical Student Wellbeing Survey, particularly the AAMC Organization of Student Representatives, as well as the members of the Better Lab including Devika Patel, Jess Hawkins, Ben Alpers, Dr. Laura Wong, and Lara Chehab for their constant support. We also thank Dr. Lee Jones (UCSF) and Adrian Anzaldua for their mentorship and guidance. We appreciate Pamela Derish (UCSF) for assistance in manuscript editing and Amy Shui (UCSF) for assistance in statistical analysis. Finally, we thank Dr. Max Feinstein (1993–2020) for the passion and joy he brought to our lives and for the beautiful memories we keep of him.

## Author Contributions

**Conceptualization:** Nikhil Rajapuram.

**Data curation:** Nikhil Rajapuram, Megan R. Marshall, Amanda Sammann.

**Formal analysis:** Nikhil Rajapuram, Simone Langness.

**Funding acquisition:** Nikhil Rajapuram, Amanda Sammann.

**Investigation:** Nikhil Rajapuram, Simone Langness, Amanda Sammann.

**Methodology:** Nikhil Rajapuram, Simone Langness, Megan R. Marshall.

**Project administration:** Nikhil Rajapuram, Simone Langness, Amanda Sammann.

**Resources:** Amanda Sammann.

**Supervision:** Simone Langness, Amanda Sammann.

**Validation:** Simone Langness.

**Writing – original draft:** Nikhil Rajapuram.

**Writing – review & editing:** Nikhil Rajapuram, Simone Langness, Megan R. Marshall, Amanda Sammann.

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
