## [Decision Letter · Decision Letter 0]

3 Sep 2020

PONE-D-20-23258

Medical Students in Distress:

The Impact of Gender, Race, Debt, and Disability

PLOS ONE

Dear Dr. Nikhil Rajapuram,

Thank you for submitting your manuscript to PLOS ONE. After careful consideration, we feel that it has merit but does not fully meet PLOS ONE’s publication criteria as it currently stands. Therefore, we invite you to submit a revised version of the manuscript that addresses the points raised during the review process.

In particular, please review the comments from Reviewer 1 and make appropriate revisions.

We look forward to receiving your revised manuscript.

Kind regards,

Yuka Kotozaki

Academic Editor

PLOS ONE

Journal Requirements:

4. We note that Figure 2 in your submission contain map images which may be copyrighted. All PLOS content is published under the Creative Commons Attribution License (CC BY 4.0), which means that the manuscript, images, and Supporting Information files will be freely available online, and any third party is permitted to access, download, copy, distribute, and use these materials in any way, even commercially, with proper attribution. For these reasons, we cannot publish previously copyrighted maps or satellite images created using proprietary data, such as Google software (Google Maps, Street View, and Earth). For more information, see our copyright guidelines: http://journals.plos.org/plosone/s/licenses-and-copyright.

4.1.    You may seek permission from the original copyright holder of Figure 2 to publish the content specifically under the CC BY 4.0 license. 

4.2.    If you are unable to obtain permission from the original copyright holder to publish these figures under the CC BY 4.0 license or if the copyright holder’s requirements are incompatible with the CC BY 4.0 license, please either i) remove the figure or ii) supply a replacement figure that complies with the CC BY 4.0 license. Please check copyright information on all replacement figures and update the figure caption with source information. If applicable, please specify in the figure caption text when a figure is similar but not identical to the original image and is therefore for illustrative purposes only.

Reviewers' comments:

Reviewer's Responses to Questions

**Comments to the Author**

1. Is the manuscript technically sound, and do the data support the conclusions?

Reviewer #1: Partly

Reviewer #2: Yes

2. Has the statistical analysis been performed appropriately and rigorously? 

Reviewer #1: Yes

Reviewer #2: Yes

3. Have the authors made all data underlying the findings in their manuscript fully available?

Reviewer #1: Yes

Reviewer #2: Yes

4. Is the manuscript presented in an intelligible fashion and written in standard English?

Reviewer #1: Yes

Reviewer #2: Yes

5. Review Comments to the Author

Reviewer #1: Thank you for the opportunity to review this paper, I found the approach and results quite interesting.

I commend the authors on their novel investigation and aim of identifying risk factors for distress. Validation of their survey, the MSWS (portions not included in the MS-WBI) was not addressed. No information about the piloting of these questions was given. A copy of the survey was not available. These are critical to understanding the validity of the assessment, which is central to the validity of the findings.

Self-selection bias is a major limitation of this study and the approach using a liaison was poorly defined, for example, how many liaisons agreed to distribute the survey?Finally, variables added to the regression model are noted in the introduction, with the exception of disability, which is not discussed until the methods and results. Having a solid framework for choosing all variables is important. I would add this reasoning to the background/introduction alongside the others.

Reviewer #2: Despite the limitations you recognize (eg. response rate), this study offers significant contributions to conversations about medical schooling conditions that deserves wide dissemination in the public sphere as well as to the medical community. Foremost, this study identifies a need for "course correction" in the ways that medical schools & the profession address student well-being. As you state: "Our data suggest that changes to the underlying structural components of medical education are likely to yield more substantial results to improve medical student well being than continued focus on individuals." This recommendation to attend to structure is critical. Further, this study reveals that debt is a significant factor in students distress. Again, this is something, as the authors' indicate, that needs to be addressed at the structural level. For future studies and/or in disseminating and explaining the significance of these findings (op-eds, medical school leadership, undergrad institutions), I would encourage the authors to look at studies addressing the effects of student debt in education (outside of medical context) (eg Hormel & McAlistair) and those that show that small amounts of debt - as little as $5,000 are an effective deterrent for students of color and under-represented students in the pursuit of graduate or first professional school (Millet). Since this study does have demographic information, I wonder if there are additional findings around what the authors point to as "demographic changes" and their findings about stress that could be highlighted. The authors do state that their data show more work is needed to understand the links between disability and documented increased stress levels that were indicated by their data. As medical schools seek to diversify, they will need to address not only recruitment and cost, but retention and completion - taking into account the importance of well-being in that equation. The "fixes," as this study suggests, cannot just be individual, but must be structural. A different kind of conversation needs to take place and this study is a very important step in that direction - to think about the increased stress and its unequal distribution (and I am guessing the unequal distribution of other causal factors like debt) on people with disabilities, women, and students of color.

6. PLOS authors have the option to publish the peer review history of their article (what does this mean?). If published, this will include your full peer review and any attached files.

Reviewer #1: No

Reviewer #2: **Yes: **Jeanne Scheper

---

## [Author Response · Author response to Decision Letter 0]

19 Sep 2020

We thank you for your time and insightful critiques about our manuscript. We appreciate the overall positive feedback and we hope to strengthen our manuscript by addressing your concerns.

1. Survey validation. In developing our survey instruments, we scoured the literature for previously validated questions that contribute to medical student wellbeing and burnout. Unfortunately, no single survey questionnaire contained what we were searching for and many wellbeing surveys were designed for either older populations, where chronic health conditions and dependence are more prevalent, or corporations, where productivity and distractions are of higher concern. As such, we opted to create a novel instrument centered primarily around the MS-WBI. Additional questions added to our survey were designed under the guidance of previously validated surveys, utilizing Likert scales and topics previously known to contribute to burnout and stress. These foundational surveys have been included as references in our manuscript in addition to a clarifying statement.

2. Survey piloting. We appreciate the reviewer addressing our oversight in not describing the piloting of the survey. The survey was piloted by 10 medical students from various institutions to ensure the survey was of adequate length, easy to comprehend and comprehensive. After an initial round of feedback regarding the survey, we made several edits and then underwent a second round of piloting, at which point there was universal acceptance of the survey for all participants. A description of this process has been included in the manuscript. A copy of the survey has been provided as supplemental material for the manuscript.

3. Medical student liaisons. We have added the total number of medical student liaisons to our manuscript.

4. Rationale for disability in the regression model. We included disability in our regression model for severe distress as disability amongst medical students has risen substantially over the last decade and the effects of disability on wellbeing was previously unknown. Our study demonstrates that identifying as having a disability is a risk factors for having severe distress. We believe that this finding contributes significantly to the literature and we hope that publishing our results will better inform medical schools and help design interventions to mitigate this risk. We have included additional information about disability amongst medical students in our introduction.

---

## [Decision Letter · Decision Letter 1]

12 Oct 2020

PONE-D-20-23258R1

Medical Students in Distress:

The Impact of Gender, Race, Debt, and Disability

PLOS ONE

Dear Dr. Nikhil Rajapuram,

Thank you for submitting your manuscript to PLOS ONE. After careful consideration, we feel that it has merit but does not fully meet PLOS ONE’s publication criteria as it currently stands. Therefore, we invite you to submit a revised version of the manuscript that addresses the points raised during the review process.

We look forward to receiving your revised manuscript.

Kind regards,

Yuka Kotozaki

Academic Editor

PLOS ONE

Reviewers' comments:

Reviewer's Responses to Questions

**Comments to the Author**

1. If the authors have adequately addressed your comments raised in a previous round of review and you feel that this manuscript is now acceptable for publication, you may indicate that here to bypass the “Comments to the Author” section, enter your conflict of interest statement in the “Confidential to Editor” section, and submit your "Accept" recommendation.

Reviewer #1: All comments have been addressed

Reviewer #2: All comments have been addressed

2. Is the manuscript technically sound, and do the data support the conclusions?

Reviewer #1: Yes

Reviewer #2: Yes

3. Has the statistical analysis been performed appropriately and rigorously? 

Reviewer #1: I Don't Know

Reviewer #2: I Don't Know

4. Have the authors made all data underlying the findings in their manuscript fully available?

Reviewer #1: Yes

Reviewer #2: Yes

5. Is the manuscript presented in an intelligible fashion and written in standard English?

Reviewer #1: Yes

Reviewer #2: Yes

6. Review Comments to the Author

Reviewer #1: The authors have done a commendable job addressing the feedback. I would suggest accepting with minor revisions.

A few notes:

While the authors are correct that their findings are novel and add to the literature, they fail to mention the disability finding in the conclusion portion of the abstract.

Consider adding more updated citations for student depression/suicidal ideation.

Puthran R, Zhang MW, Tam WW, Ho RC. Prevalence of depression amongst medical students: A meta‐analysis. Medical education. 2016 Apr;50(4):456-68.

Rotenstein LS, Ramos MA, Torre M, Segal JB, Peluso MJ, Guille C, Sen S, Mata DA. Prevalence of depression, depressive symptoms, and suicidal ideation among medical students: a systematic review and meta-analysis. Jama. 2016 Dec 6;316(21):2214-36.

The authors should add disability status to this paragraph under novel questions:

The validated MS-WBI questionnaire was administered within the survey with permission from the authors. Novel questions were created based on foundations established from previously validated wellbeing survey methods [27-29]. Changes in students’ physical, emotional, and social domains of wellbeing since entering medical school were assessed on a five-point Likert scale. Additionally, respondents were asked about their frequency of participation in activities known to be protective against distress and burnout, such as exercise, meditation/mindfulness practices, healthy eating, hobby participation, and connecting with a loved one [30-33]. Frequency of actual participation in protective activities was compared against students’ self-reported ideal participation frequency.

I’m not sure this needs to be included (could be removed for word count)

We first contacted the Association of American Medical Colleges (AAMC). However, the AAMC administers a national survey that targets second-year and graduating medical students only, does not include the MS-WBI and would not allow for individual response tracking. We also contacted several Medical School Deans who voiced concern that respondents were asked to identify their medical school.

In this paragraph I would mention that these numbers are higher than the latest findings (the implication being that students may not feel protected/well supported in disclosing to their schools.

We found that nearly 10% of survey respondents identified as having a disability and that

disability status nearly doubled the risk of severe distress (OR 1.81). Self-reporting of disability

in medical school has risen substantially over the last three years, specifically for psychological

and chronic health issues [41]. Little is currently known about the additional stressors faced by

students with disability and to what degree accommodations from medical schools are being

offered and utilized. Understanding this topic more thoroughly may prove pivotal for guiding

314 medical schools in designing interventions to support this vulnerable group.

Reviewer #2: (No Response)

7. PLOS authors have the option to publish the peer review history of their article (what does this mean?). If published, this will include your full peer review and any attached files.

Reviewer #1: No

Reviewer #2: No

---

## [Author Response · Author response to Decision Letter 1]

1 Nov 2020

We thank you for your time and insightful critiques about our manuscript. We appreciate the overall positive feedback and we hope to strengthen our manuscript by addressing your concerns.

1. Abstract Correction. We appreciate the reviewer addressing our oversight in not including disability in the conclusion of the abstract. We have added disability status as a novel risk factor to the abstract conclusion.

2. Updated Citations. We have included updated citations on student depression/suicidal ideation as indicated.

3. Disability/Chronic Illness Status. We have added the fact that one of our novel questions was asking about disability/chronic illness status. We also included that a higher number of students self-identified as having a disability than has been found in prior literature possibly due to student discomfort.

---

## [Decision Letter · Decision Letter 2]

18 Nov 2020

Medical Students in Distress:

The Impact of Gender, Race, Debt, and Disability

PONE-D-20-23258R2

Dear Dr. Nikhil Rajapuram,

We’re pleased to inform you that your manuscript has been judged scientifically suitable for publication and will be formally accepted for publication once it meets all outstanding technical requirements.

Kind regards,

Yuka Kotozaki

Academic Editor

PLOS ONE

Additional Editor Comments (optional):

Reviewers' comments:

Reviewer's Responses to Questions

**Comments to the Author**

1. If the authors have adequately addressed your comments raised in a previous round of review and you feel that this manuscript is now acceptable for publication, you may indicate that here to bypass the “Comments to the Author” section, enter your conflict of interest statement in the “Confidential to Editor” section, and submit your "Accept" recommendation.

Reviewer #1: All comments have been addressed

Reviewer #2: All comments have been addressed

2. Is the manuscript technically sound, and do the data support the conclusions?

Reviewer #1: Yes

Reviewer #2: Yes

3. Has the statistical analysis been performed appropriately and rigorously? 

Reviewer #1: I Don't Know

Reviewer #2: I Don't Know

4. Have the authors made all data underlying the findings in their manuscript fully available?

Reviewer #1: Yes

Reviewer #2: Yes

5. Is the manuscript presented in an intelligible fashion and written in standard English?

Reviewer #1: Yes

Reviewer #2: Yes

6. Review Comments to the Author

Reviewer #1: The revisions to this article strengthen the authors presentation and sufficiently address the prior feedback. The authors should be commended for this work and the novel addition of disability status to the discussion of wellbeing.

Reviewer #2: Author's had previously addressed my concerns and recent revisions were to address the concerns of other readers. While the study has limitations (not a large or perfect sample), it still has sufficient data to highlight an important and neglected area. This research will stimulate further work, hopefully, to map and address these concerns.

7. PLOS authors have the option to publish the peer review history of their article (what does this mean?). If published, this will include your full peer review and any attached files.

Reviewer #1: No

Reviewer #2: No

---

## [Editor Report · Acceptance letter]

25 Nov 2020

PONE-D-20-23258R2 

Medical students in distress:The impact of gender, race, debt, and disability 

Dear Dr. Rajapuram:

I'm pleased to inform you that your manuscript has been deemed suitable for publication in PLOS ONE. Congratulations! Your manuscript is now with our production department. 

Kind regards, 

on behalf of

Dr. Yuka Kotozaki 

Academic Editor

PLOS ONE